# Degradation of Praguicide Disulfoton Using Nanocompost and Evaluation of Toxicological Effects

**DOI:** 10.3390/ijerph20010786

**Published:** 2022-12-31

**Authors:** Mayne Veronesi, Mariandry Rodriguez, Grazielle Marinho, Cleide Aparecida Bomfeti, Bruno Alves Rocha, Fernando Barbosa, Marília Cristina Oliveira Souza, Márcia Cristina da Silva Faria, Jairo Lisboa Rodrigues

**Affiliations:** 1Instituto de Ciência, Engenharia e Tecnologia, Universidade Federal dos Vales do Jequitinhonha e Mucuri (UFVJM), Campus Mucuri, Teófilo Otoni 39803-371, MG, Brazil; 2Analytical and System Toxicology Laboratory, Department of Clinical Analyses, Toxicology and Food Sciences, School of Pharmaceutical Sciences of Ribeirao Preto, University of Sao Paulo, Avenida do Cafe s/no, Ribeirao Preto 14040-903, SP, Brazil

**Keywords:** pesticides, degradation, remediation, toxicity, organophosphates, biomonitoring

## Abstract

Organophosphates (OPPs) are an important element of modern agriculture; however, because they are being used excessively, their residues are leaching and accumulating in the soil and groundwater, contaminating aquatic and terrestrial food chains. An important OPP called disulfoton is frequently used to eradicate pests from a wide range of crops, including Brazil’s coffee crops. Additionally, it does not easily degrade in the environment, and as such, this compound can slowly build up in living organisms such as humans. Moreover, this compound has been classified as “extremely hazardous” by the World Health Organization. This study evaluated the degradation efficiency of disulfoton using a Fenton-like reaction catalyzed by magnetite nanoparticles and determined the toxicity of the by-products of the degradation process using the bioindicator *Allium cepa*. Further, the removal efficiency of disulfoton was determined to be 94% under optimal conditions. On the other hand, the *Allium cepa* bioassay showed different toxic, cytotoxic, genotoxic, and mutagenic outcomes even after the remediation process. In conclusion, the Fenton process catalyzed by magnetite nanoparticles presents great efficiency for the oxidation of disulfoton. However, it is important to highlight that the high degradation efficiency of the Fenton-based process was not sufficient to achieve detoxification of the samples.

## 1. Introduction

The contamination by pesticides in the environment has gradually increased and is harming living beings. The toxicity of environmental pollutants can affect the ecosystem and human health since the vast majority of them can potentially bioaccumulate in aquatic flora and fauna, allowing their entry into the food web [1].

Since 2008, Brazil has been the largest consumer of pesticides in the world. The impacts of pesticides on public health are diverse, reaching vast territories and involving different population groups, such as workers in different fields of activity, laborers at factories and farms, and consumers who eat contaminated food. The impacts are associated with the current development model, which is primarily focused on the production of goods for export [2].

The expansion of the number of chemical substances listed in the Draft Resolution of the Chamber of Deputies (RPC) no. 5, on the 28^th^ of September 2017, Annex XX [3] can lead to the naturalization of contamination, consequently rendering it banal, as if this form of pollution were legalized [2]. The quality criteria for water for human consumption reflect, over time, growing pollution from various processes, including the agricultural process of using dozens of pesticides and chemical fertilizers. In addition, from January to March 2019, the Ministry of Agriculture, Livestock, and Supply (MAPA) approved 121 new pesticide registrations. For example, Act No. 17 of the Department of Plant Health and Agricultural Inputs, published in the Diary da Union on March 21^st^, 2019, granted registrations to 35 new labels, which are now licensed to be sold and consumed throughout the national territory. Among the approved compounds, six are classified as class I, which comprises compounds considered “extremely toxic” to human health [4].

Bioaccumulation in the food web is possible and can constitute a risk or hazard to animals and people over time. It was reported that pesticides cause up to three million cases of acute and severe poisoning annually, with as many or more cases unreported, and approximately 220,000 deaths globally [5].

Additionally, organophosphate pesticides (OPPs) are widely used around the world; these compounds are very toxic when absorbed by the human body, as they cause an influx of the enzyme acetylcholinesterase [6]. Organophosphates are used in agriculture as insecticides because they are highly efficient against pests, have low bioaccumulation, and rapidly degrade in the environment by projection and hydrolysis [7].

Further, beginning with the concern about environmental effects and the contamination of water resources, food safety, and public health, a method of degrading the contaminant disulfoton, belonging to the organophosphate chemical group, was developed. Advanced oxidative processes were used with magnetite nanoparticles. The degradation process was monitored by gas chromatography–mass spectrometry. In addition, cytotoxic, genotoxic, and mutagenic effects were evaluated with the bioindicator *Allium cepa*, which is considered excellent for such testing due to its sensitivity and ease of analysis in environmental studies [8].

## 2. Materials and Methods

All chemicals were of analytical grade and were used as received without any further purification. The high-purity deionized water (resistivity 18.2 MΩ cm) used throughout the experiment was obtained using a Milli-Q water purification system (Millipore RiOs-DITM, Bedford, MA, USA). Analytical standard solutions of disulfoton (O, O-diethyl S-2-ethylthioethyl phosphorodithioate) and the internal standard, anthracene, were purchased from Sigma-Aldrich (St. Louis, MO, USA). Two solvents (HPLC grade) were used in the dispersive liquid–liquid microextraction procedure: acetonitrile and hexadecane. H_2_O_2_ was used in the Fenton reaction.

### 2.1. Summary of Magnetite (Fe_3_O_4_) Nanoparticles

The Fe_3_O_4_ nanoparticles were synthesized according to the method in which preparation occurs by co-precipitation of iron (Fe^2+^) and iron (Fe^3+^) ions in an aqueous NaOH solution. In addition, ~5 g of NH_4_Fe(SO_4_)_2_·12H_2_O and ~8 g of (NH_4_)_2_Fe(SO_4_)_2_·6H_2_O were dissolved in 100 mL of water. After that, Fe_3_O_4_ was precipitated at room temperature using 100 mL of 2 M NaOH under agitation for 30 min. The precipitate was washed multiple times with water and dried at room temperature in a vacuum desiccator.

### 2.2. Degradation of Disulfoton Using Magnetite Nanoparticles

The disulfoton degradation method using magnetite nanoparticles in this study was previously described [9]. In this study, modifications were proposed for optimal pesticide remediation. Consequently, an experimental design was created in order to optimize the degradation procedure and determine the effect of various factors on the process (Table 1 and Table 2). Disulfoton samples were prepared in 100 mL volumetric flasks with concentrations of 2, 10, and 50 µg.L^−1^ and transferred to polypropylene tubes (10 mL).

The amounts of nanoparticles added to the tubes with the compound ranged from 200 to 600 mg. The pH of samples was adjusted (5, 7, and 9) using HCl (0.1 M) or NaOH (0.1 M). After pH adjustment, different amounts of hydrogen peroxide (200, 400, or 600 µL) were added to each sample. The samples were mixed on an orbital shaker at 300 rpm for 15 min, then centrifuged for 5 min, and a magnet was used to separate the nanoparticles from the sample.

### 2.3. Sample Preparation

The microextraction methodology was adapted from Zanjani et al. [6], using a disulfoton solution (purchased from PerkinElmer, New York, NY, USA). The standard solution was prepared in acetone (purchased from ISOFAR, Sao Paulo, Brazil) at a concentration of 20,000 µg L^−1^. Additionally, the concentrations of 50, 10, and 2 µg L^−1^ were used to perform monitoring via gas chromatography.

The extraction of organophosphate compounds from water samples was carried out according to the dispersive liquid–liquid microextraction (DLLME) method. A mass of 0.5 g of NaCl and 10 μL of the internal standard (anthracene) were added to each reaction tube. Further, a mixture of 180 μL of hexadecane (the extraction solvent) and 420 μL of acetonitrile (the dispersive solvent) were rapidly added for compound extraction. After that, the samples were centrifuged for 5 min and placed in a polystyrene box with ice for 10 min. The organic phase was collected and transferred to an Eppendorf tube. The organic phase was evaporated, and the dried residue was dissolved with 100 µL of acetonitrile, vortexed for 10 s, and injected into the gas chromatograph for GC-MS.

### 2.4. Instrumentation and Sample Analysis

The analyses were performed using a gas chromatography system coupled to mass spectrometry (GC-MS) from Thermo Fisher Scientific^®^, Waltham, MA, USA. The determination of disulfoton was performed using an SLB^®^-5 ms analytical column (30 m × 0.25 mm × 0.25 µm; Sigma-Aldrich^®^). The initial temperature of the oven was 80 °C, which was maintained for 2 min, followed by a heating ramp of 15 °C/min up to 300 °C, which was maintained for 5 min, for a total of 21.67 min running time. Additionally, helium was used as a carrier gas at a flow rate of 1 mL/min. The injector temperature was 230 °C, the injection volume was 1 µL, the procedure was performed in splitless mode, and the valve was opened after 1 min.

The detection system used was a quadrupole-type mass spectrometer (ISQ single quadrupole model, Thermo Fisher Scientific^®^) equipped with an electron impact ionization source. The temperatures of the ionization source and the transfer line for the mass spectrometer were maintained at 300 and 290 °C, respectively.

Furthermore, the determinations were performed in full scan mode, monitoring the range of mass/load ratio (*m*/*z*) from 80 to 280, and in selected ion monitoring mode (SIM) with selected *m*/*z* values of 274–89–88 (quantification). The data acquisition start time was 8 min. Data acquisition and quantification were performed using Thermo Xcalibur^TM^ version 2.2 (Thermo Fisher Scientific^®^).

The method was validated through tests of precision, accuracy, linearity, robustness, and the limits of detection and quantification.

### 2.5. Evaluation of Toxicity Using Allium Test

The experiment was carried out with adaptations according to [10,11,12]. Three parameters were evaluated: cyto-, geno-, and mutagenicity. The negative control (C−) was carried out with water, and the positive control (C+) with methyl methanesulfonate (MMS), 10 ppm. A number of 30 *Allium cepa* seeds were put on filter paper in a Petri dish, hydrated with water, and allowed to germinate until the roots reached 1 cm in length. The obtained seedlings were then exposed to water prepared with disulfoton at concentrations of 2, 10, and 50 µg.L^−1^ before and after degradation and to water prepared with 200 mg magnetite. Moreover, all the samples were prepared in triplicate and subjected to germination for 24 h. After germination, the roots were cut and fixed using Carnoy’s solution (ethanol:acetic acid, 3:1) in 1.5 mL microtubes for 24 h. Carnoy’s solution was then replaced with 1 mL of 70% alcohol, and the samples were maintained for 24 h. The volume of 70% ethanol was replaced, while the roots were stored in the refrigerator until slide preparation.

For slide preparation, the roots were washed three times with water. After that, they were subjected to acid hydrolysis with a 1 M HCl solution at 60 °C in water and kept in a water bath for 9 min. After acid hydrolysis, the roots were transferred to microtubes coated with aluminum foil containing Schiff’s reagent (Merck, Rahway, NJ, USA) for 2 h.

Additionally, the tip of each root was cut and placed on a blade, and a drop of 2% acetic carmin was added. After 9 min, the root tip was covered with a coverslip and carefully macerated for microscopic analysis.

## 3. Results and Discussion

### 3.1. Degradation of Disulfoton

The efficacy of the Fenton reaction in the degradation of disulfoton was studied. In this reaction, hydroxyl radicals formed by the hydrolysis of H_2_O_2_ in the presence of magnetite nanoparticles oxidized the organic pesticide present in the sample [9]. By using the initial concentration (Ci) and residual concentration (Cr) of each peak after gas chromatography analysis, it was feasible to determine the degradation rate (D) of each component, as described in Equation (1):D = (Ci − Cr)/Ci × 100(1)

Several parameters affect the Fenton process and interfere with the degradation of organic compounds, such as pH, volume of H_2_O_2_, nanoparticle mass of Fe_3_O_4_, and concentration of analytes; the Fenton’s reagent and pH are the most influential factors [13]. In order to optimize the degradation process, a 24 factorial design was prepared, complete without a central point, in which the parameters concentration, pH, volume of H_2_O_2_, and weight of nanomaterial in the degraded contaminant disulfoton were evaluated. Table 1 and Table 2 show the studied parameters and their respective levels, as well as the development of factor planning, and Table 3 shows the degradation rates.

#### Effects of Disulfoton Degradation Variables

The results of the effects, errors, and time for the variables analyzed in the design are presented in Table 4. Given the results obtained, it appears that the concentration did not have an influence; however, the volume effect, pH, and weight of the nanomaterial positively influenced the degradation of disulfoton at the levels studied. The second-order effects, however, did not influence the rate of degradation.

Figure 1 shows the Pareto diagram of the standardized effects at *p* = 0.05. The absolute value of the impact of each variable and their interaction is given to the right of each bar. Further, all the results to the right of the dashed line with values larger than 2.571 (*p* = 0.05) are significant for the degradation of the compound disulfoton. Figure 1 shows that three of the main effects were significant (pH, volume, and weight), showing that these factors influenced the degradation rate. After analyzing the second-order effects, it was seen that none of the interactions were significant.

According to Pignatello [15], the pH of the medium plays a fundamental role in the efficiency of the Fenton and photo-Fenton processes. The values above 3.0 cause Fe (III) to precipitate in the form of insoluble hydroxide, and at values below 2.5, high concentrations of H^+^ can sequester hydroxyl radicals. There is also a predominance of less hydroxylated species that have less absorptivity, with the need for pH control being the biggest limitation of these processes. However, no significant differences were observed in the samples due to the change in pH.

A study by Verma and Haritash [16] analyzing the role of pH, FeSO_4_, and H_2_O_2_ in the degradation of the antibiotic amoxicillin found that an excess of H_2_O_2_ in the Fenton reaction caused the sequestration of hydroxyl and hydroperoxyl radicals, which have less potential for reduction, thus hindering the degradation process.

In a study by Watts et al. [17], the results showed that in the presence of magnetite, the contaminant pentachlorophenol was totally degraded in 12 h, while in the presence of hematite, only 12% was degraded, showing the greater activity of magnetite. The authors noted that magnetite was more efficient in degrading compounds because it consists of Fe^2+^ and Fe^3+^, which have the greatest potential to catalyze the decomposition of hydrogen peroxide.

In a study by Chen et al. [14], the formation of hydroxyl radicals increased due to the addition of iron ions in a constant amount, and this factor had more influence than other operational parameters. This has also been observed in studies related to the degradation of lignin by the Fenton reaction.

Figure 2, Figure 3 and Figure 4 graphically represent the second-order effects. Although no interaction showed a significant influence, the interaction of H_2_O_2_ volume and pH indicates that with increased volume and acidification, there was an increase in the degradation rate, and this effect also occurred when there was a high nanoparticle weight and slight acidification, when both interactions showed a degradation rate close to 95%. Additionally, the interaction between volume and nanoparticles was also favorable, but it presents degradation rates below 90%.

### 3.2. Evaluation of Toxicological Potential by Allium cepa Bioassay

#### 3.2.1. Analysis of Cytotoxic Potential

The statistical analysis of the results of the factorial design suggests that the rate of degradation of the compound is mainly related to the pH and the amount of nanomaterial. Therefore, according to the results of the degradation rate and the significant effects analyzed, the best conditions for disulfoton degradation are as follows: H_2_O_2_ at a volume of 600 μL, magnetite weight of 600 mg, and a pH of 5. Under these conditions, both concentrations are calculated as:MI = (N^o^. of cells in division)/(N^o^. of cells analyzed) × 100(2)

While evaluating meristematic cells of *A. cepa* in the process of cell division, it was possible to determine the mitotic index (MI) for each group. The values found are listed in Table 5.

An analysis of the positive control (MMS) was performed in order to assess the sensitivity of the test, along with evaluating the negative control and samples.

In the analysis of variance, Student’s *t*-test, and non-parametric Mann–Whitney test, it was found that the experimental procedure was effective, with significant values at the level of 5%.

The samples containing the contaminant disulfoton had a significant reduction in the mitotic index (MI) when compared to the negative control, revealing that these samples were contaminated by cytotoxic substances.

The results of the microscopic analysis referring to cell divisions are shown in Table 5. The data of each sample were analyzed using GraphPad Prism version 9 software, student version, to verify the behavioral relationships between samples.

The results obtained for the mitotic index show a significant correlation (*p* < 0.05) between the negative control and the samples contaminated with pesticide, according to the Mann–Whitney test, with a value of *p* ≤ 0.0001. The sample containing only magnetite showed no statistically significant difference when compared to the negative control, with a value of *p* = 0.4693. Thus, the results obtained by means of statistical tests indicate that the samples containing disulfoton, both before and after degradation, had cytotoxic potential at the microscopic level.

Cellular progress can be blocked when certain external stimuli are present, such as exposure to pesticides; this action is called mitoinhibition. Any deviation from the ordered and directed progression of the cell cycle is reflected in a state of cytotoxicity and genotoxicity. Mitogens act to overcome the braking mechanisms that block the progression of the cell cycle, and their action is called myostimulation [18].

Figure 4 shows the results for the mean and standard deviation of the analyzed samples. The effect of disulfoton consisted of an increased interphase index, and as the concentration increased, cell division was inhibited at different stages of mitosis.

Pesticides are bioactive molecules that can form metabolites, and due to their electrophilic characteristics, they are able to react with and combine with biomolecules such as DNA and induce changes [19].

It can also be noted that the samples showed the greatest reductions in MI when subjected to the degradation process using a greater number of nanoparticles. This can be explained by the results of a study by Popescu et al. [20], which showed that pure Fe_3_O_4_ is significantly viable for all investigated periods of exposure, as the nanoparticles of magnetite have been shown to be biocompatible; however, there is reduced efficiency and increased toxicity when combined with gemcitabine (GEM), which is a cytostatic drug used in the treatment of several types of cancer.

Nanostructuring with GEM showed a cytotoxic effect, with viability reduced by 60% compared to the control. The different concentrations and exposure times were evaluated, and it was determined after the treatment that viability fell below 80% at 0.12 mg/mL and below 40% at 0.15 mg/mL. Thus, the viability of nanoparticles relative to that of the drug is dependent on dose and time. It was reported that the loading of GEM into magnetic chitosan nanoparticles increased the cytotoxic effect of the drug in breast cancer cells after treatment; viability was reduced below 40%, making it possible to evaluate the highly cytotoxic effect on the systems [20,21].

#### 3.2.2. Analysis of Genotoxic Potential

The chromosomal aberrations evaluated for genotoxicity analysis are characterized by changes in chromosome structures at different stages of cell division, which can occur spontaneously or as a result of exposure to contaminants.

This genotoxicity study was performed by analyzing the chromosomal aberrations (CAs) present in the slides prepared for the study. The alterations considered in this study were binucleation, adherence, bridge, C-metaphase, necrosis, and loss and breakage in all phases of cell division; CAs present in about 5000 cells from each sample were evaluated for genotoxicity (Figure 5).

The results regarding chromosomal aberrations are shown in Table 6.

According to the results obtained through microscopic analysis to assess genotoxicity and subsequent statistical analysis using the Mann–Whitney test, it can be said that the analyzed samples had a genotoxic effect, as they presented statistically significant differences when compared to the negative control.

The positive control and the sample containing only magnetite showed a *p*-value of 0.0024 and 0.0023, respectively, whereas samples contaminated with disulfoton showed a *p* ≤ 0.0001.

Silvério et al. [22] evaluated pesticide exposure in rural workers in southern Minas Gerais, where disulfoton is the most commonly used organophosphate, representing almost 70% of the application in the region.

According to the authors, exposure to organophosphates can cause acute and chronic effects, because their results showed that compared to the control group, workers exposed to organophosphates had a significantly higher incidence of muscle tremors, weakness, irritability, restlessness, blurred vision, dizziness, tingling of limbs, abdominal cramps, nausea, breathing difficulties, nasal irritation, increased bronchial secretions, cough, decreased hearing, and tinnitus; in addition, budding changes and condensed chromatin and karyolitic cells were found when the cytoma of the oral mucosa was used as a biomarker for genotoxicity.

Several types of chromosomal aberrations were observed in the present study (Table 6 and Figure 6). This indicates that DNA damage cannot be easily repaired [23].

Thus, the occurrence of various types of aberrations in the chromosomes of meristematic cells of *A. cepa* roots can be attributed to the collective effect of clastogenic and aneugenic actions by various contaminating compounds [24].

According to Seth et al. [25] the induction of breaks, chromosomal losses, and damage to DNA in plants indicates that the contaminant being tested by the bioindicator has clastogenic potential.

DNA damage can be associated with the generation of free radicals, causing DNA strand breaks and irreversible damage, by binding to proteins involved in DNA replication, repair, recombination, and transcription [26,27].

It can be seen that necrosis was the most frequent chromosomal aberration in the analyzed samples, which can be explained by the response of cells to irreversible damage to the genetic material. Different mechanisms for maintaining cell homeostasis are linked to the processes of cell division, cell metabolism, and cell death [28].

The failure of the repair mechanism can promote neoplastic transformation, indicating the beginning of the tumor formation process. Thus, unrepaired cells are sent to the cell death process (known as necroptosis in pathology) or programmed cell necrosis in order to prevent the onset of tumors [29].

Cell necrosis involves the decomposition of negatively affected cell groups [30], where mitochondria increase in volume and dry up with little or no need for energy. Further, there is no synthesis of protein or nucleic acid, nor does new gene transcription occur, and DNA is digested randomly.

Necrosis is characterized by the early disappearance of ion pumping activities due to damage to the membrane or depletion of cellular energy [31].

#### 3.2.3. Analysis of Mutagenic Potential

The analysis of mutagenic potential was performed based on the frequency of micronuclei (MN) in meristematic cells in all phases of the cell cycle. For this purpose, 5000 cells from each sample were analyzed to obtain the mutagenic potential and the frequency index of MN through the relationship between dividing cells and the presence of MN.

The micronuclei found in the analyses were present in cells in interphase and prophase, as can be seen in Figure 7. The data obtained through microscopic analysis are shown in Table 7.

According to the results obtained through microscopic analysis to assess mutagenicity and the subsequent statistical analysis, it can be said that the analyzed samples had a mutagenic effect, as they presented a statistically significant difference when compared to the negative control.

The positive control, the sample containing only magnetite, and the samples contaminated with disulfoton showed values of *p* ≤ 0.0001.

Organochlorine and organophosphate pesticides are the most common compounds produced by the pesticide industry worldwide [32]. Various effluents from industrial wastewater and sludge have shown high mutagenic potential [33]. Several scientific studies on the genotoxicity of wastewater and pesticides have suggested a direct association with the mutagenicity of pollutants in water bodies and possible risks to human health [34,35,36,37]. 

According to Santos [34], organophosphates represent a class of pesticides with high toxicity in widespread use, for which it is difficult to assess the effects of long-term exposure and low doses due to the absence of clinical manifestations [35].

The exposure to these compounds causes damage to the nervous and respiratory systems and the reproductive organs, dysfunctions in the immune and endocrine systems, as well as mutagenicity and carcinogenicity [36]. Further, the biomonitoring of workers exposed to pesticides indicated the occurrence of MNs, while the formation of MNs is induced by substances that cause chromosomes (clastogens) to break. Micronuclei are chromosomal fragments or whole chromosomes that are not included in the nucleus during cell division, forming a much smaller nucleus [37]. Many cancers have an epithelial origin, suggesting that micronuclei in epithelial cells represent an important biomarker that can be used in epidemiological studies [38]. Figure 8 shows the means and standard deviations of the micronuclei found in the analyzed samples.

## 4. Conclusions

In this study, the results show that the Fenton process catalyzed by magnetite nanoparticles described here presents great efficiency in the oxidation of the compound disulfoton, since the obtained results demonstrate that there was up to 94% degradation under optimal conditions, established after factorial planning aimed at optimizing the methodology.

Furthermore, the results of the present study show that there was induction of cell division, formation of MN, and a significant increase in chromosomal aberrations, confirming the cytotoxic, genotoxic, and mutagenic potential of disulfoton.

The evaluation of cytotoxicity, genotoxicity, and mutagenicity demonstrated that the samples had toxicological potential, even after degradation of the compound and reduced concentration. These findings clearly indicate that the indiscriminate use of the organophosphate pesticide disulfoton is highly dangerous. In view of the common practice of applying this compound on agricultural land, it is necessary to develop methodologies for environmental remediation in areas contaminated with this type of compound, as pollutants can enter the food chain and cause risks to human health.

## Figures and Tables

**Figure 1 ijerph-20-00786-f001:**
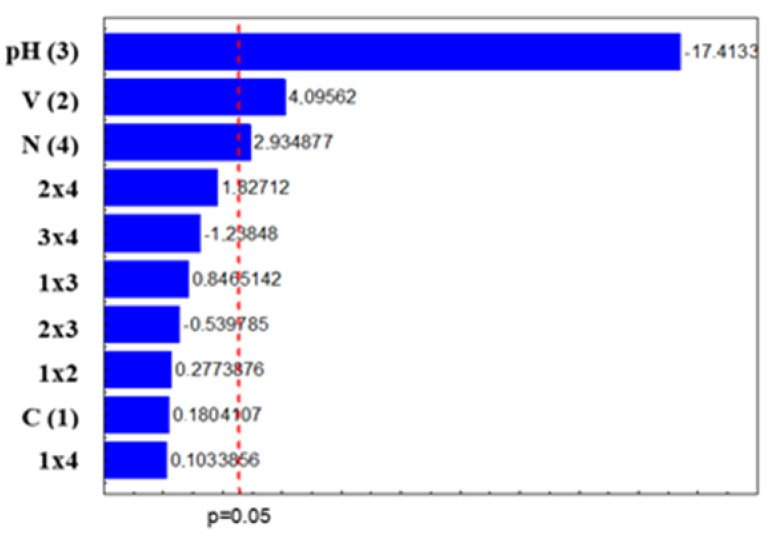
Pareto graph of standardized effects at *p* = 0.05 for disulfoton degradation. It can be seen that pH had the greatest influence on the degradation rate. In [14], it was found that the reaction is favorable in an acid medium with a pH between 3 and 5 for degradation by the Fenton process.

**Figure 2 ijerph-20-00786-f002:**
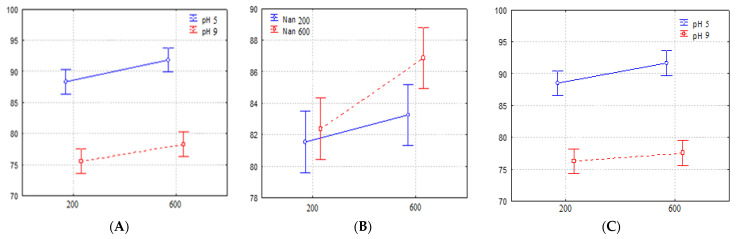
Effects of (**A**) second-order/volume × pH interaction; (**B**) second-order/volume × nanoparticle interaction; and (**C**) second-order/nanoparticle × pH interaction.

**Figure 3 ijerph-20-00786-f003:**
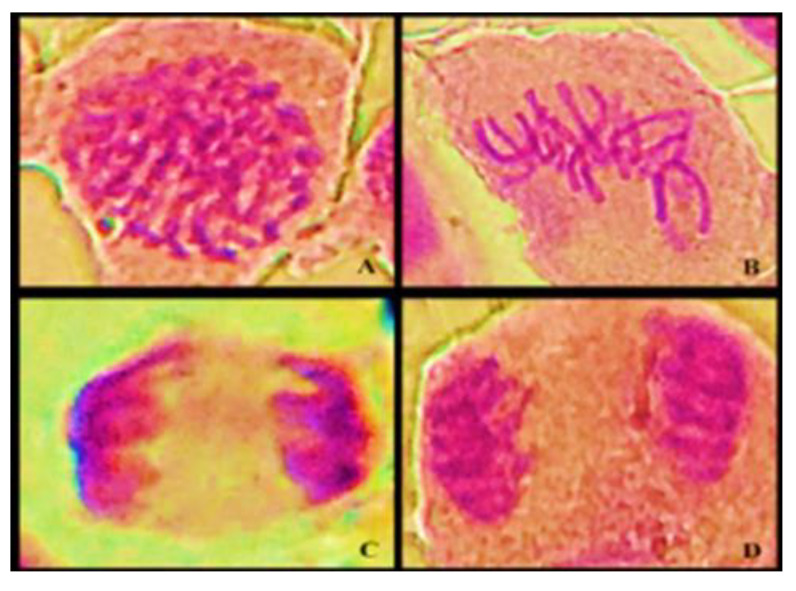
A light microscope photomicrograph of cell division phases: (**A**) prophase, (**B**) metaphase, (**C**) anaphase, and (**D**) telophase.

**Figure 4 ijerph-20-00786-f004:**
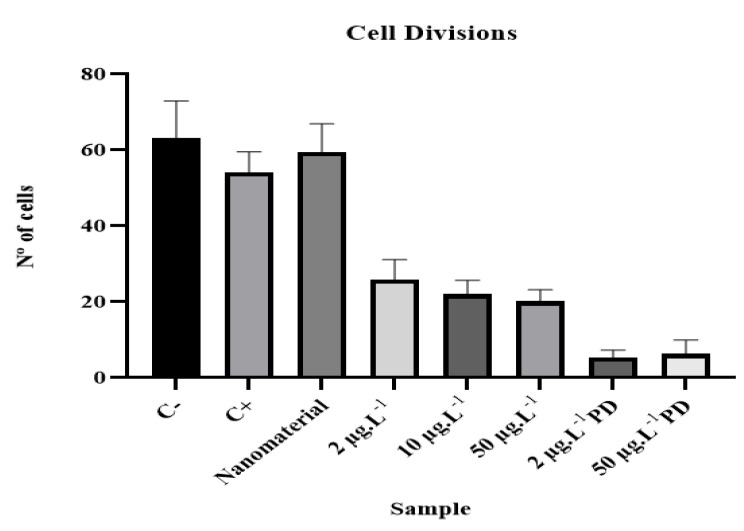
Average values and standard deviations in cell division of analyzed samples.

**Figure 5 ijerph-20-00786-f005:**
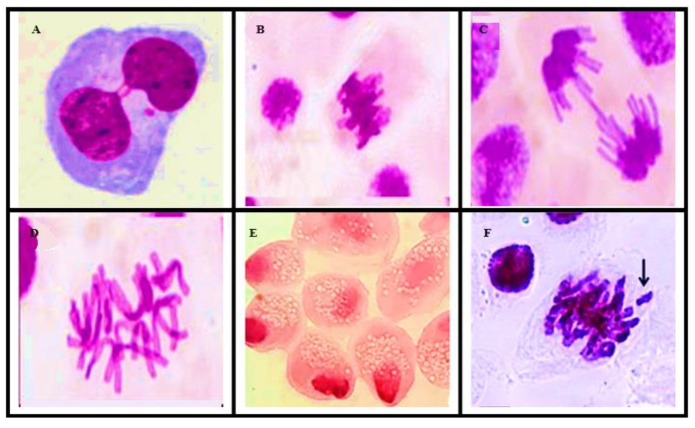
Optical microscope photomicrographs of chromosomal aberrations: (**A**) binucleation, (**B**) metaphase with adherence, (**C**) anaphase with bridge, (**D**) C-metaphase, (**E**) necrosis, and (**F**) chromosomal loss.

**Figure 6 ijerph-20-00786-f006:**
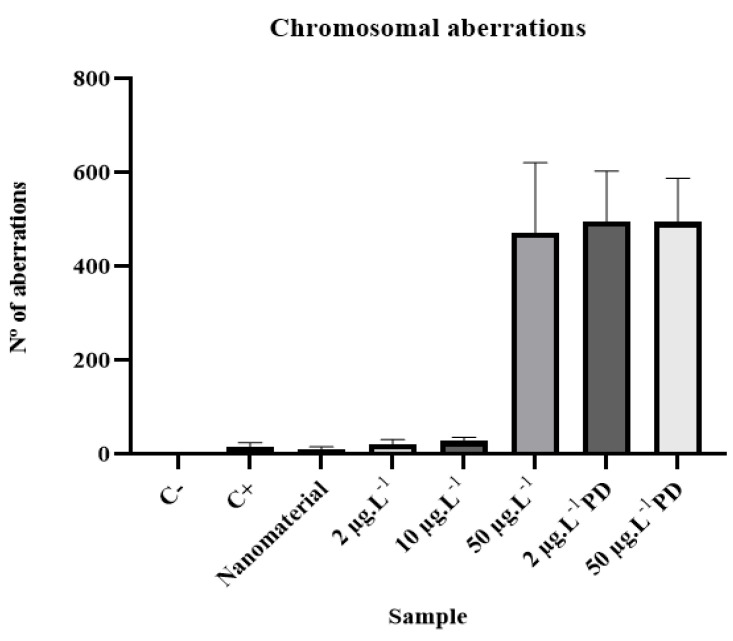
Mean values and standard deviations for chromosomal aberrations of analyzed samples.

**Figure 7 ijerph-20-00786-f007:**
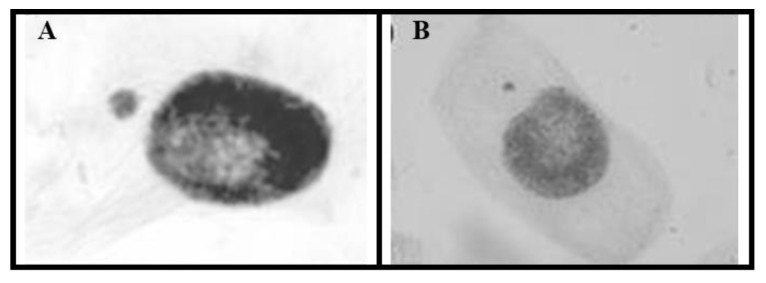
Light microscope photomicrographs of micronuclei analyzed in mitotic phases: (**A**) MN in interphase and (**B**) MN in prophase.

**Figure 8 ijerph-20-00786-f008:**
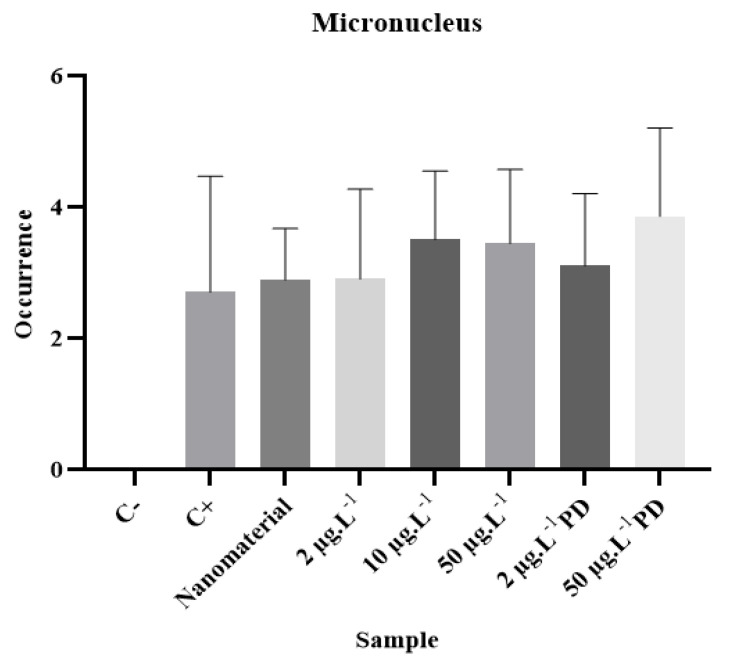
Average values and standard deviations of micronuclei from analyzed samples.

**Table 1 ijerph-20-00786-t001:** Experimental domain for a 2^4^ factorial experimental design.

Factors (X)	Levels		
	−1	0	1
Disulfoton (μg L^−1^/X_1_)	2	10	50
Hydrogen peroxide (μL/X_2_)	200	400	600
pH (X_3_)	5	7	9
Magnetite nanoparticles (mg/X_4_)	200	400	600

**Table 2 ijerph-20-00786-t002:** Complete factorial planning study.

Experiment	X_1_	X_2_	X_3_	X_4_
**1**	−1	−1	−1	−1
**2**	1	−1	−1	−1
**3**	−1	1	−1	−1
**4**	1	1	−1	−1
**5**	−1	−1	1	−1
**6**	1	−1	1	−1
**7**	−1	1	1	−1
**8**	1	1	1	−1
**9**	−1	−1	−1	1
**10**	1	−1	−1	1
**11**	−1	1	−1	1
**12**	1	1	−1	1
**13**	−1	−1	1	1
**14**	1	−1	1	1
**15**	−1	1	1	1
**16**	1	1	1	1
**17**	0	0	0	0
**18**	0	0	0	0
**19**	0	0	0	0
**20**	0	0	0	0

**Table 3 ijerph-20-00786-t003:** Disulfoton compound degradation rate (%).

Sample	Ci	Cr	%
**1**	2.09	0.27	87
**2**	49.76	6.12	88
**3**	2.09	0.20	90
**4**	49.76	5.47	89
**5**	2.09	0.48	77
**6**	49.76	12.67	75
**7**	2.09	0.52	75
**8**	49.76	10.72	78
**9**	2.09	0.21	90
**10**	49.76	5.49	89
**11**	2.09	0.12	94
**12**	49.76	3.16	94
**13**	2.09	0.54	74
**14**	49.76	11.70	76
**15**	2.09	0.42	80
**16**	49.76	10.06	80
**17**	10.58	1.48	86
**18**	10.58	1.27	88
**19**	10.58	1.27	88
**20**	10.58	1.38	87

**Table 4 ijerph-20-00786-t004:** Effects and errors of factors studied in a full 2^4^ factorial design for mining.

Factor	Estimated Effect ± Error (0.756)	t
Mean	83.5	221.0
Concentration	0.14	0.2
Volume	**3.10**	**4.1**
pH	**−13.16**	**−17.4**
Weight	**2.21**	**2.9**
1 × 2	0.21	0.3
1 × 3	0.64	0.8
1 × 4	0.08	0.1
2 × 3	−0.41	−0.5
2 × 4	1.38	1.8
3 × 4	−0.93	−1.2

**Table 5 ijerph-20-00786-t005:** Number of dividing cells and mitotic index.

Sample	N^o^. of Cells Analyzed	N^o^. of Cells in Interphase	N^o^. of Cells in Mitosis	Mitotic Index (MI)
C− *	5000	4370	630	12.60
C+ **	5000	4460	540	10.80
Magnetite	5000	4406	594	11.88
2 µg L^−1^	5000	4742	258	5.08
10 µg L^−1^	5000	4781	219	4.38
50 µg L^−1^	5000	4799	201	4.02
2 µg L^−1^ PD	5000	4948	52	1.04
50 µg L^−1^ PD	5000	4938	65	1.30

* Ultrapure water; ** MMS; PD, post-degradation.

**Table 6 ijerph-20-00786-t006:** Chromosomal aberrations found in analysis.

Sample	Binucleation	Adhesion	Bridge	C-Metaphase	Necrosis	Break	Total CA
C− *	-	-	2	-	-	2	4
C+ **	28	14	40	19	-	27	141
Magnetita	32	-	34	-	-	11	91
2 µg L^−1^	25	16	9	14	123	19	213
10 µg L^−1^	45	9	7	-	201	3	271
50 µg L^−1^	63	13	15	11	4600	7	4717
2 µg L^−1^ PD	-	-	-	-	4948	-	4948
50 µg L^−1^ PD	-	-	-	-	4938	-	4938

* Ultrapure water; ** MMS; PD, post-degradation.

**Table 7 ijerph-20-00786-t007:** Results of micronuclei found in analyses.

Sample	No. of Cells Analyzed	No. of Micronuclei	Frequency MN (%)
C− *	5000	-	-
C+ **	5000	27	0.60
Magnetita	5000	26	0.59
2 µg L^−1^	5000	29	0.61
10 µg L^−1^	5000	21	0.44
50 µg L^−1^	5000	31	0.65
2 µg L^−1^ PD	5000	31	0.63
50 µg L^−1^ PD	5000	27	0.55

* Negative control. ** Positive control.

## Data Availability

The datasets generated during and/or analyzed during the current study are available from Jairo Lisboa upon reasonable request.

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
