# Peer review of "Degradation of Praguicide Disulfoton Using Nanocompost and Evaluation of Toxicological Effects"

_ijerph, 2022, doi:10.3390/ijerph20010786_

Round 1

Reviewer 1 Report

The paper manuscript “Degradation of Praguicide Disulfoton Using Nanocompost and Evaluation of Toxicological Effects.”

Overall, although the study could have a potential to be accepted, the authors should fully revise the manuscript and significally improve the quality of presentation especially in the context of the results as well as to extend the introduction and the conclusion.

In addition, some comments follow.

1)General Comment: Please check abbreviations with consistency in main text. Define it at the first appearance, then use it after the definition (e.g. OPPS, DLLME, MMS, MI, etc.).

2) Line 46: Please remove “…he ….”

3) Lines 49-55: The sentence should be rewritten and splited in further sentences.

4)Line 66-67: The sentence should be rephrased.

5) Lines 76-83: The sentence should be rewritten and splited in further sentences.

6) To replace “μg.L-1” to  μg L-1”.

7) Lines 118-122: The paragraph to be rewritten. To replace the phrase “author”.

8) Line 135: the phrase “which contains the chromatographic column” could be removed.

9) Is the GC-MS method fully validated?

10) Line 149: Please replace “YES”  to “SIM”.

11)Line 147 To be checked the reported m/z values.

12) Line 151 To be corrected typo errors (e.g.1, -12, -13).

13) Tables 3 & 4 to be revised.

14) More clear figures to be provided.

15) Line 272: Sentence to be moved above Figure 3.

16) To adjust  and harmonise references according to journal format.

Author Response

We would like to thank you for the critical analysis of our manuscriptthat certainly contributed to improve the quality of the manuscript.

Below are the answers:

1)General Comment: Please check abbreviations with consistency in main text. Define it at the first appearance, then use it after the definition (e.g. OPPS, DLLME, MMS, MI, etc.).

I agree. The alteration was done.

2) Line 46: Please remove “…he ….”

I agree. Removed

3) Lines 49-55: The sentence should be rewritten and splited in further sentences.

I agree. The alteration was done.

4)Line 66-67: The sentence should be rephrased.

I agree. The alteration was done.

5) Lines 76-83: The sentence should be rewritten and splited in further sentences.

I agree. The alteration was done.

6) To replace “μg.L-1” to  “μg L-1”.

I agree. The alteration was done.

7) Lines 118-122: The paragraph to be rewritten. To replace the phrase “author”.

I agree. The alteration was done.

8) Line 135: the phrase “which contains the chromatographic column” could be removed.

I agree. The alteration was done.

9) Is the GC-MS method fully validated?

Inserted in the text, line 146

10) Line 149: Please replace “YES”  to “SIM”.

I agree. The alteration was done.

11)Line 147 To be checked the reported m/z values.

I agree. The alteration was done.

12) Line 151 To be corrected typo errors (e.g.1, -12, -13).

I agree. The alteration was done.

13) Tables 3 & 4 to be revised.

I agree. The alteration was done.

14) More clear figures to be provided.

I agree. The alteration was done.

15) Line 272: Sentence to be moved above Figure 3.

I agree. The alteration was done.

16) To adjust  and harmonise references according to journal format.

I agree. The alteration was done.

Reviewer 2 Report

The paper need a final review and there are some problem with the chemical formulas that need attention. The temperatures in centigrade degrees (xxx ºC) are not weel formatted.

All the references session must be review because the automatic formation  worked  properly. The way the author are written in not coherent along the session.

Author Response

The paper need a final review and there are some problem with the chemical formulas that need attention. The temperatures in centigrade degrees (xxx ºC) are not weel formatted.

Answer: I agree. We made the correction.

All the references session must be review because the automatic formation  worked  properly. The way the author are written in not coherent along the session.

Answer: I agree. We made the correction.

The English revision and correction service offered by the journal was used.

Reviewer 3 Report

The current structure of the manuscript is not suitable for publication as it has a lot of errors, especially in formatting which questions both the integrity and quality of the manuscript. 

Author Response

The current structure of the manuscript is not suitable for publication as it has a lot of errors, especially in formatting which questions both the integrity and quality of the manuscript. 

Answer:  We made the correction. A general review of the formatting and structures of the manuscript was carried out. The English revision and correction service offered by the journal was used.

Round 2

Reviewer 3 Report

The paper of Veronesi et al. aimed to evaluate the degradation capacity of the organophosphate contaminant in water by a Fenton-Like reaction catalyzed by magnetite nano particles. The authors also used bioassay to assess its toxicological effects. Overall, the paper has met the objectives of the paper. I have some comments which you can see directly in the manuscript (See attached). There are some technical errors in the manuscript. For example, affiliations were not mentioned. In the references, there are number "1" across all references. Please fix these things. 

Author Response

We would like to thank you very much for considering our manuscript for publication as well as for sending us the second round of reviewer’s comments. The peer-review process provided us excellent suggestions and directions, which has greatly enhanced the quality of our presentation. We would like to inform you that the suggestions made the reviewer were taken into account and we have revised the manuscript carefully, including references and English language (MDPI Language Editing Services). 

Below you will find the list of clarifications to review comments. We hope that the revised manuscript is suitable for publication in International Journal of Environmental Research and Public Health. All the corrections and revisions are presented with the “track-changes” of Microsoft Word in manuscript and in supplementary information.  

Sincerely,

Prof. Dr. Jairo Lisboa Rodrigues. .

Instituto de Ciência, Engenharia e Tecnologia, Universidade Federal dos Vales do  Jequitinhonha e Mucuri (UFVJM), Campus Mucuri, Teófilo Otoni, MG, Brazil. E-mail: [email protected]

Reviewer #1: Referee Report to IJERPH-2024532

1) General comments: The paper of Veronesi et al. aimed to evaluate the degradation capacity of the organophosphate contaminant in water by a Fenton-Like reaction catalyzed by magnetite nano particles. The authors also used bioassay to assess its toxicological effects. Overall, the paper has met the objectives of the paper. I have some comments which you can see directly in the manuscript (See attached). There are some technical errors in the manuscript. For example, affiliations were not mentioned. In the references, there are number "1" across all references. Please fix these things:

Answer:  We would like to thank reviewer #1 for the critical analysis of our manuscript that certainly contributed to improve the quality of the manuscript. We took into consideration the suggestions of reviewer #1 and we addressed all of his comments. Moreover, the manuscript was completely revised, some alterations were made to clarify all questions and enhance the quality of this study.

2) Specific comments: Affiliation was not included.

Answer:  The requested information was included.

3) Specific comments: cite the works of Reyes et al. 

Reyes, V. P., Ventura, M. A., & Amarillo, P. B. (2022). Ecotoxicological Assessment of Water and Sediment in Areas of Taal Lake with Heavy Aquaculture Practices Using Allium cepa and Daphnia magna Assay. Philippine Journal of Science, 151(3), 969–974.

Answer:  The reference was included.

4) Specific comments: Figure 5. loss of what?

Answer: Thanks for your comment. The correct sentence is “Chromosome loss”.

5) Specific comments: Format the references.

Answer:  The manuscript was carefully checked by all authors and the corrections were made when necessary. Moreover, the reference section will be formatted by MDPI editorial.